# Fabrication and Characterization of Hollow Polysiloxane Microsphere Polymer Matrix Composites with Improved Energy Absorption

Sofia Gabriela Gomez * , Andrea Irigoyen, Stephanie Gonzalez, Kevin Estala-Rodriguez, Evgeny Shafirovich , Md Sahid Hassan , Saqlain Zaman and Yirong Lin

Department of Aerospace and Mechanical Engineering, The University of Texas at El Paso, El Paso, TX 79968, USA
* Correspondence: sggomez@miners.utep.edu

**Abstract:** Hollow polymer microspheres with superior elastic properties, high thermal stability, and energy absorbance capabilities are essential in many applications where shock and vibration need to be mitigated, such as in civil, medical, and defense industries. In this paper, the synthesis, fabrication, and characterization of hollow thermoset microspheres for syntactic polymer foam were studied. The hollow polymer microspheres (HPMs) were made by developing core–shell composites and thermally removing the polystyrene core to yield a polysiloxane shell. The HPMs were embedded into a polydimethylsiloxane (PDMS) matrix to form a polymer syntactic foam. The mechanical energy absorption characteristic of polymer syntactic foams was measured by cyclic uniaxial compression testing following ASTM 575. The engineered compression response was demonstrated by fabricating and testing syntactic foams with different porosities, ranging from a 50 vol% to 70 vol% of HPMs. Through scanning electron microscopy (SEM), we observed that the HPM contributes to the energy absorption of the syntactic foam. Moreover, Fourier transform infrared spectroscopy (FTIR) and thermogravimetric analysis (TGA) determined the necessity of a profound study to understand the effects of varying HPM synthesis parameters, as well as the syntactic foam fabrication methods. It was shown that the compressive modulus and toughness can be increased by 20% using a 70 vol% of porosity with synthesized HPM syntactic foams over bulk PDMS. We also found that the energy absorbed increased by 540% when using a 50 vol% of porosity with fabricated HPM-PDMS syntactic foams.

**Keywords:** hollow polysiloxane microspheres; syntactic foam; polydimethylsiloxane; core–shell composites

## 1. Introduction

Hollow microspheres have been widely studied because of their broad applications, such as pharmaceutical [1], biological [2], energy [3], defense [4], and cosmetics [5] applications. Microspheres have the capability of lowering the density of materials they are incorporated in. When added to a matrix, the matrix material will weigh less than the same product without the hollow spheres, which is ideal for many applications, such as those where lightweight characteristics, thermal isolation, and impact absorption are required. Microsphere's physical characteristics make them extremely effective insulators [6] that are capable of reflecting and dissipating heat [7] while also being fire resistant [8]. Polymer hollow microspheres are polymeric spheres that have a size range from nanometers to microns. The hollow microspheres are generally made from thermoplastics, are usually hydrophobic, and have high binding capabilities [9]. Recently, more attention has been brought to polymer hollow microspheres as they have a large specific surface area, relatively low density, high encapsulation capabilities, and thermal insulation properties [10]. Those with high surface areas and excellent electrical properties have exhibited outstanding microwave absorption capabilities [11]. An advantage of hollow microspheres is their

ability to be customized in terms of density and diameter. This helps reduce the weight of products while maintaining high mechanical strength. Hollow polymer particles have been actively investigated due to their surface functionality, good flowability, optical properties, and surface permeability [12]. Microspheres not only have these characteristics capable of improving the mechanical behavior of 3D printed products, but they are also ideal candidates for shape memory applications that require structural complexity [13].

Previously, hollow polymer microspheres were created using the swelling method [14], liquid droplets, the dried-gel droplet method, self-assembly methods, microencapsulation, emulsion polymerization, and the template method [15]. However, these are associated with expensive equipment, high operation costs, and tedious methodologies. Hollow spheres have generated significant interest, and the goal is to fabricate them faster to have as many of them as possible to utilize. One of the simplest methods is to coat the core with a shell and subsequently remove the core [16]. Polysiloxane has outstanding temperature and oxidative stability, excellent low-temperature flexibility, and high resistance to weathering and chemicals, making polysiloxane a desirable shell material. In this work, hollow polysiloxane microspheres were synthesized through polymerization that utilized hydrochloric acid (HCL) and methyltrimethoxysilane (MTMS), which created a shell on the polystyrene particles. The core can be removed in several ways, such as dissolution, chemical etching, or, as in this work, calcination. The development of hollow polymer microspheres has promoted their increased usage in integrations involving syntactic foam fabrication. Syntactic foams are porous composites that are found in various applications and fields, such as in the marine, aerospace, defense, and energy sectors [17]. Polymer matrix composites can also be designed to feature high levels of energy dissipation, elasticity, and low density [18]. Recently, polymer composites have become popular in aircraft applications as they present properties that can be used for structural weight reduction and high compressibility rates [19]. The high compression rates of syntactic foams are advantageous for foams that undergo heavy loading. They are fabricated by dissipating hollow particles in a polymer matrix. The closed cell structure allows for the matrix to obtain higher mechanical properties.

Fabrication methods and the material characterization of polymer syntactic foams have been explored in previous research. Yousaf et al. investigated the compression properties of polymeric syntactic foam composites under cyclic loading [20]. The syntactic foams were fabricated by adding hollow microspheres of various sizes and wall thicknesses into a polyurethane matrix. Researchers analyzed stress–strain curves, which revealed the viscoelastic properties of the material. SEM images also exhibited high elastic recovery and energy dissipation over compressional strain testing. Moreover, high volume fraction and wall thickness particles in syntactic foams have enabled such parameters to be independent of each other, thus allowing mechanical properties to be tailored. As such, Gupta et al. researched a collection of polymer matrix syntactic foams, ranging from those composed of metal to polymer matrices. The polymer syntactic foam study showed that the macroscopic thermal conductive properties are directly correlated to its microscopic characteristics, such as hollow microsphere wall thickness and volume percentage [21]. The importance of polymer composite structures has been expressed by structural and defense industries. Madenci et al. discussed how pultruded fiber-reinforced polymer (PFRP) material has high strength and chemical resistance that makes it desirable in the replacement of metal materials [22]. They found that one of the PFRP samples studied had an increased load capacity of about 130%, which was due to the orientation of web openings and reinforcement materials. Previous studies have shown that the microscopic properties of polymeric composite structures have a direct impact on their macroscopic integrity. We were motivated to explore the production and refinement of polymer syntactic foams to examine their potential implementation in scenarios that require high compressive strength while maintaining a low density. This work presents experiments that could offer direct benefits to practical applications such as aerospace sandwich structures [23], biomedical devices [24], and composite cross-arms used in the power industry [25].

In this study, through the modification of core–shell ratios, attempts at incorporating comonomers and plasticizers, and variation in calcination temperatures, polysiloxane hollow spheres were fabricated. Monodisperse and polydisperse hollow thermoset microsphere systems were integrated into a PDMS matrix at a 50 vol% and 70 vol% to study the effects of varying microstructures on compressive reliability. Damage tolerance, compressive response, Young's modulus, and energy absorption were analyzed by performing cyclic compression testing of the syntactic foam specimens. The morphology of the hollow polysiloxane microspheres and the syntactic foam's microstructure were characterized through scanning electron microscopy (SEM). Fourier transform infrared spectroscopy (FTIR) and thermogravimetric analysis (TGA) aided in the investigation of attaining and confirming the hollowness of polysiloxane thermoset shells, as well as in the thermal degradation of the polystyrene in the core–shell polymer composites. Throughout this work, it was shown that the performance of syntactic foams was highly dependent and directly correlated to its microstructure; therefore, we methodically optimized the polymerization, calcination, and handling stages to obtain hollow polymer microspheres with integral mechanical properties.

## 2. Materials and Methods

### 2.1. Polystyrene Core Synthesis

The synthesis of the polystyrene cores was induced using dispersion polymerization, as shown in Figure 1. To polymerize polystyrene, a nitrogen purge is necessary to prevent oxidation in the system that inhibits free-radical polymerization [26]. The monomer used for the polystyrene core synthesis process was styrene (St, 99% extra pure, ACROS Organics, Geel, Belgium) [27]. A five-neck round-bottomed flask with a solution of 2-methoxyethanol (MeCell, 99+% extra pure, ACROS Organics) and ethanol (EtOH, 200 proof, Pharmco, WI, USA) was used as the reaction medium because of its solubility with styrene [28]. To initiate radical polymerization, benzoyl peroxide (BPO, EMD Millipore Corporation, Burlington, MA, USA) was added after the addition of styrene into the solution [29]. To stabilize the system, hydroxypropyl cellulose (HPC, Mw = 100,000 g/mol, ACROS Organics) was incorporated into the mixture [30]. The round-bottomed flask containing the chemicals for polymerization was submerged in an ethylene glycol bath that was heated by a hot plate at a reaction temperature of 75 °C while being magnetically stirred at 500 RPM. After 24 h, the polystyrene particles were washed in reagent alcohol. A condenser was mounted to the system to equilibrate the reaction temperature and a thermocouple was also placed in another spout (Figure 1A). The resulting solution underwent centrifugal forces at 4000 RPM for 15 min to separate unpolymerized styrene and reaction medium from polystyrene powder. The cleaning of polystyrene powder was conducted through repetitive centrifuging and vortex. Reagent alcohol was used each time until the separation of materials was distinct, which was indicated by a clear solution on the top and white polystyrene powder on the bottom (Figure 1B). The solution was discarded, and the polystyrene was left to dry overnight in a petri dish. After drying, a mortar was used forcefully on the polystyrene powder to deagglomerate the system and yield individual polystyrene cores Figure 1C).

### 2.2. Polystyrene–Polysiloxane Core–Shell Composite Fabrication

Sol–gel polymerization was the method used to coat the polystyrene cores with a polysiloxane shell [16]. Trimethoxymethylsilane (MTMS, Sigma-Aldrich, St. Louis, MO, USA) was used as the polysiloxane crosslinker. The hydrolysis step in the sol–gel method was induced by exposing the trimethoxymethylsilane to an acidic environment using an aqueous hydrochloric acid (HCl, Fisher Scientific, Waltham, MA, USA) solution (Figure 2A). After hydrolysis, sonication induced vibrational entropy to deagglomerate the particles (Figure 2B). The condensation reaction in the sol–gel method was prompted by introducing ammonium hydroxide (NH₄OH, Sigma-Aldrich) into the beaker (Figure 2C). Another sonication step was implemented to further break up agglomerates within the sys-

tem (Figure 2D). The polymerization reaction was completed after 5 h and the composites were washed using the same procedure as the polystyrene cores (Figure 2E).

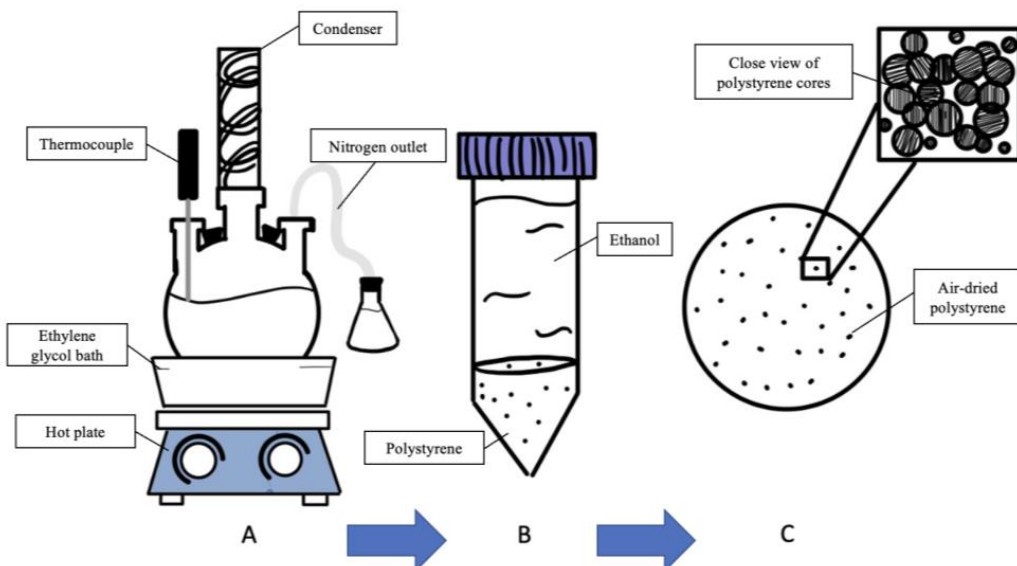

**Figure 1.** The dispersion polymerization process to synthesize polystyrene cores through (**A**) the use of a nitrogen-purged system at a desired reaction temperature, (**B**) post-processing by washing un-polymerized styrene, and (**C**) drying polystyrene powder for observation and core–shell production.

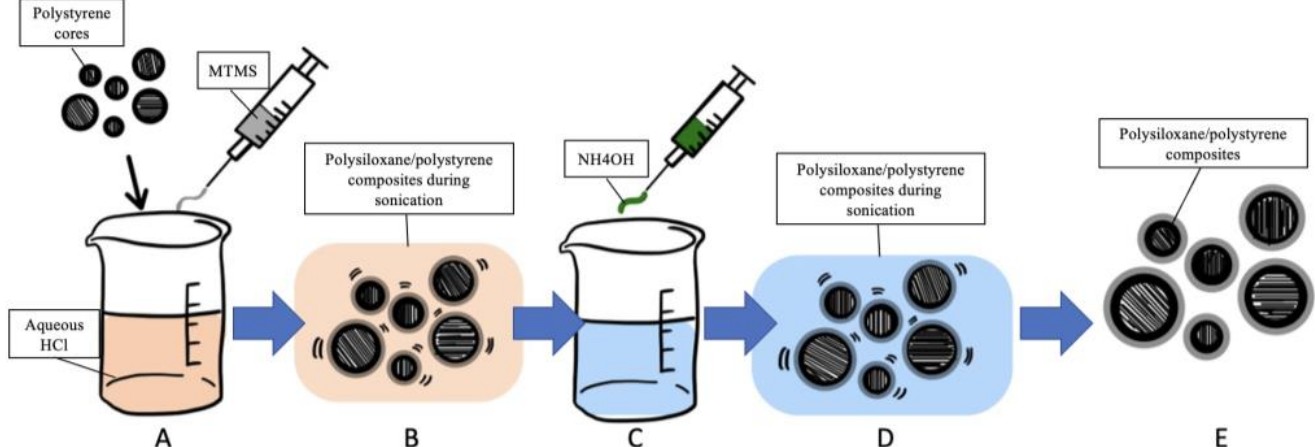

**Figure 2.** Core–shell fabrication using the sol–gel method involving (**A**) a hydrolysis stage followed by (**B**) a ten-minute sonication period. The (**C**) condensation stage is followed by another (**D**) ten-minute sonication period. (**E**) The system is left for 5 h to polymerize the shell and form the polystyrene–polysiloxane composite.

### 2.3. Hollow Polysiloxane Microsphere Generation

A calcination method was used for degrading polystyrene cores at 1 °C/min and dwelling for 30 min at 500 °C in an ambient atmosphere [31]. Since the degradation temperature of polystyrene starts at 250 °C [32] and the degradation of polysiloxane starts at 450 °C [33], we quantified different calcination temperatures and durations to understand the amount of polystyrene left in the polysiloxane shell. To fully decompose polystyrene and effectively remove carbon remnants from the polystyrene core, partial degradation of polysiloxane was required.

### 2.4. Polymer Syntactic Foam Formation

Sylgard 184 (the PDMS prepolymer and curing agent) was acquired from Dow Corning. The PDMS prepolymer was prepared as recommended in a 10:1 weight ratio with the respective catalyst. Subsequently, 4 g of PDMS base and 0.4 g of PDMS curing agent were mixed in a THINKY mixer at 2000 RPM for 4 min [34]. Hollow particles were added to the silicon elastomer and mixed in the THINKY mixer at 700 RPM for 4 min. The final mixture of PDMS precursor and hollow particles was dispensed in a compression mold made from ABS. ABS molds were additively manufactured using a Creality3d Ender 3 printer. The printer used Fused Deposition Modeling (FDM) technology to heat filament and print accurate compression molds in compliance with the ASTM 575. All PDMS specimens (porous PDMS 1, porous PDMS 2, bulk, and syntactic foams) were cured at 120 °C for 90 min. The temperature regime was selected because Sylgard 184 is effectively cured at 100 °C based on industry specifications [35]. Both temperature and duration of cure time were increased because the porous PDMS specimens were fabricated with a pore former that was etched at this temperature for the specified duration.

### 3. Characterization

The size distribution and morphology of the polystyrene cores, the core–shell polymer composites, the HPMs, and the syntactic foams were analyzed using scanning electron microscopy (SEM) on a ThermoFisher Scientific Phenom ProX Desktop SEM microscope under 15 kV accelerating voltage. The particle samples observed in SEM were prepared by carefully depositing deagglomerated powder onto carbon tape located on top of the sample holder. The excess powder was blown away with compressed air to avoid drifting. The syntactic foam composites were cut into 10 mm squares and pressed onto the carbon tape on top of the sample stage. A laser diffraction particle size analyzer (Microtrac Bluewave) was used to quantify the size distribution of the system. The resulting weight percentage of polystyrene content in the polysiloxane shell after calcination was observed in a nitrogen atmosphere using the TGA 55 from TA instruments. Residual polystyrene remnants were examined using a Nicolet iS5 FTIR spectrometer from ThermoFisher Scientific. The mechanical compression response was measured by an Instron 5568 load frame using a 10 kN load cell, and samples were tested at 50% and 70% compression of their original height. The mechanical behavior of a 50 vol% of hollow polysiloxane syntactic foams was characterized through uniaxial cyclic compression testing. In accordance with ASTM575, three test specimens of the same kind were compressed at a ramping speed of 12 mm/min until a compressive displacement of 8.3 mm was reached, measuring a 70% compression response. Direct comparisons were made with bulk PDMS compression samples and porous PDMS without hollow particle reinforcements. For cyclic compression tests, each specimen type was tested individually, and cycles were performed within 5 min of each other. To prevent material failure, a compressive strain of 0.5 mm/mm was exerted on all specimens to prevent fracture under higher strain.

### 4. Results and Discussion

#### 4.1. Polystyrene Cores

Figure 3 displays the morphology and particle size distribution of polymerized styrene that was synthesized using three different reaction temperatures. For this work, we analyzed the difference in particle synthesis between three trials: Trial A at a 75 °C reaction temperature, Trial B at a 72 °C reaction temperature, and Trial C at an 80 °C reaction temperature. Figure 3A displays that Trial A yielded the largest average diameter of polystyrene particles at 11 μm and exhibited a narrow passing % curve, suggesting that reaction temperatures at 75 °C result in a relatively stable polymerization process. A slight increase in temperature of 5 °C to a reaction temperature of 80 °C resulted in a highly varied system and a reduction in average diameter by 80% (Figure 3C). Reductions in particle size and a highly polydisperse system were observed in Trial C, which may have been the result of a more rapid free-radical polymerization process that resulted in the production of smaller

particles [36]. In Trial B (Figure 3B), a decrease in the system's temperature to 72 °C resulted in an average particle size of about 8 μm, signifying that free radicals were produced more slowly and that particle growth was able to take place to increase overall particle diameter. The effects of temperature on particle size suggest that there is a delicate threshold (within a temperature difference of only 5 °C) where styrene synthesis can result in monodisperse or polydisperse systems while also maintaining an average diameter that is larger than 2 μm.

### 4.2. Core–Shell Composites

Sol–gel polymerization was the method used to coat the polystyrene cores with a polysiloxane shell to form core–shell composites [37]. This method was conducted under ambient conditions; however, as with polystyrene synthesis, many factors contributed to the composition of the polystyrene–polysiloxane composites shown in Figure 4 Varying hydrolysis pH values (Figure 4A), no sonication (Figure 4B), and changes in PS to MTMS weight ratio (Figure 4C) all impacted the morphology of the resultant core–shell composites. The suspended particles' electrostatic equilibrium is determined by the pH of the aqueous HCl solution shown in Figure 4A. Subjecting the particles to a pH of 5.5 triggers agglomerations, which may be due to weak repulsive forces between the composites, thus suggesting that their surface charges were closer to that of the system's isoelectric point. The change in hydrolysis pH from 5.5 to 5.0 resulted in uniformity and stability, where the electrostatic repulsive forces were stronger at the slipping plane, known as the colloidal system's zeta potential [38]. The sonication process played a key role in the distribution of silanols across the polystyrene surface during the hydrolysis stage. As seen in Figure 4B, the absence of additional entropy resulted in large, agglomerated polystyrene particles within a coating of MTMS and further polymerized polysiloxane. Polysiloxane submicron spheres were generated when using a core–shell weight ratio of 1:2, as shown in Figure 4C. The excess amount of MTMS resulted in individual spheres, primarily due to polystyrene cores being already coated. Figure 4D displays the ideal composition and morphology of the polystyrene–polysiloxane composites. In this case, a hydrolysis pH value of 5.0 was used, sonication was introduced, and a 2:1 core–shell ratio was used to avoid excess MTMS. Yellow arrows are used to define areas where polysiloxane was coated on the core. The polysiloxane coating resulted in groupings of polystyrene particles that were later deagglomerated using a mortar and a pestle.

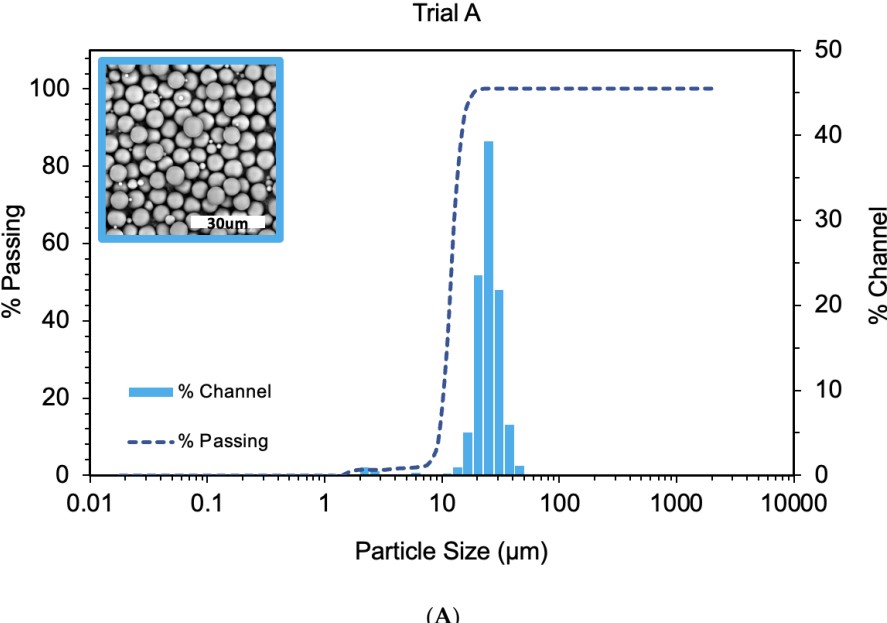

(**A**)

**Figure 3.** *Cont.*

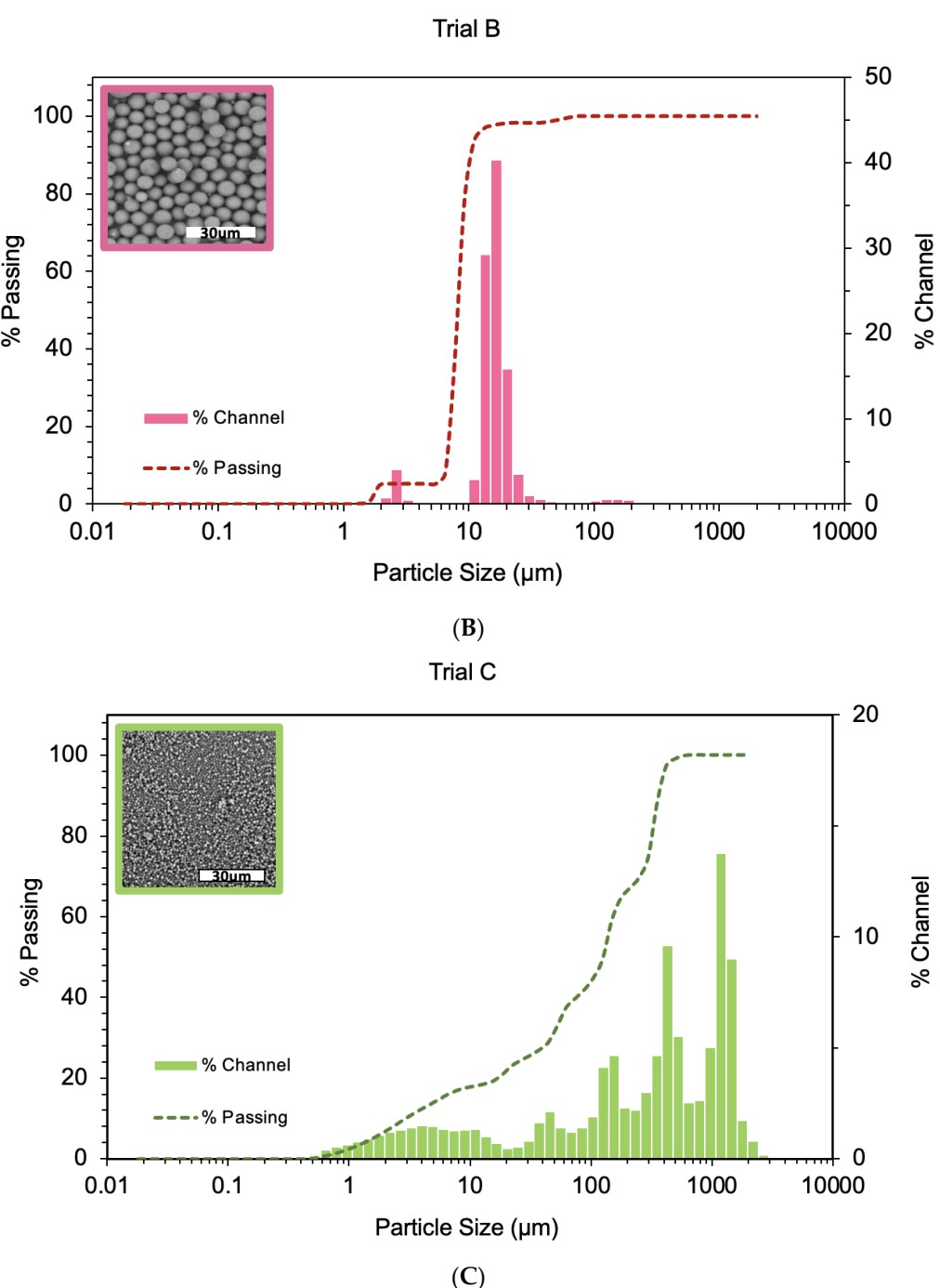

**Figure 3.** Particle size distribution and SEM images of trial polystyrene cores polymerized at (**A**) 75 °C, (**B**) 72 °C, and (**C**) 80 °C reaction temperatures.

*4.3. Polysiloxane Hollow Microspheres*

Calcination was used to remove the polystyrene core and generate hollow polysiloxane microspheres. TGA was used to determine the degradation of the previously synthesized polystyrene cores (Figure 5). As shown in Figure 5, the thermal decomposition of polystyrene begins at 250 °C and the polystyrene is effectively burnt out completely before reaching 500 °C. Using the TGA results as shown in Figure 5, the calcination protocol for the core–shell composites was determined to be above 250 °C to degrade the polystyrene cores. However, a calcination protocol that enables partial degradation of the polysiloxane shell should be considered to facilitate a physical means of removal to successfully draw out the inner polystyrene core.

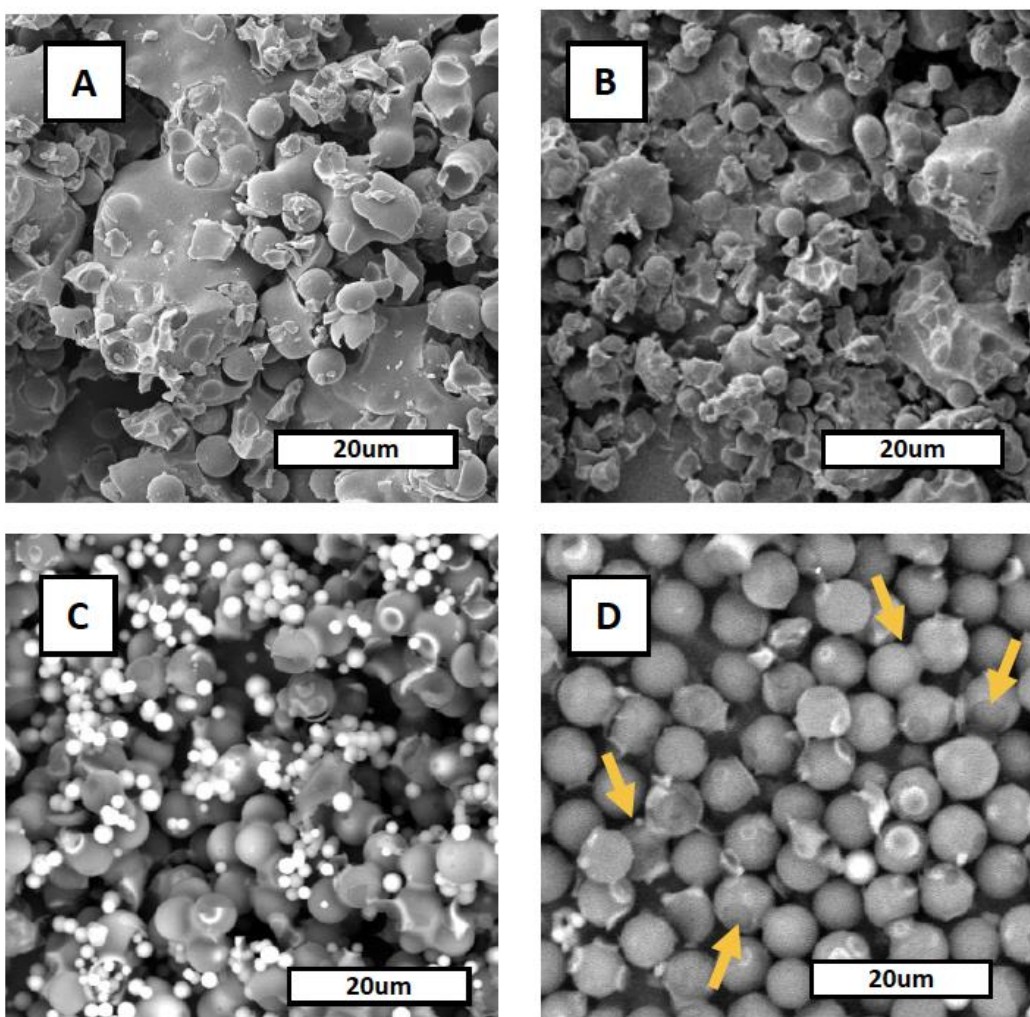

**Figure 4.** Polystyrene–polysiloxane core–shell composite fabrication under (**A**) a hydrolysis pH of 5.5, (**B**) no sonication and a hydrolysis pH of 5.0, (**C**) a 1:2 core–shell weight ratio with sonication, and (**D**) a 2:1 core–shell weight ratio with sonication and a hydrolysis pH of 5.0.

Figure 6 depicts the effects of using different calcination protocols (by varying calcination temperature or duration) on the final appearance of HPMs. Three calcination trials were conducted to determine the optimum calcination protocol for effectively withdrawing polystyrene from the polysiloxane shell. In Figure 6, Trial A refers to the first calcination protocol, which involved a ramp rate of 1 °C/min, until it reached a maximum temperature of 500 °C, and dwelled for 1 min. Trial A subjected the composites to heat treatment for a total of 16 h. Trial B refers to the calcination protocol with a ramp rate of 1 °C/min, until it reached a maximum temperature of 400 °C, and dwelled for 1 h. Trial B subjected the composites to an average of 17 h of treatment. Finally, Trial C refers to the calcination protocol with a ramp rate of 4 °C/min, until it reached a maximum temperature of 500 °C, and dwelled for 1 h. Trial C subjected the composites to an average of 5 h of treatment. Slower ramp rates were chosen to reduce thermal shock to the polysiloxane, which could have resulted in possible breakage or increased brittleness. Dwell temperatures of 400 °C and 500 °C were chosen because the degradation temperature of polystyrene is between 250 and 400 °C. Trial B was the only protocol where the composites were calcinated at 400 °C, and a higher weight percentage loss was observed. Weight loss in the temperature range of 400–500 °C indicated that polystyrene was still present inside the shell and was etched only when the polysiloxane was partially degraded. The residual polystyrene resulted in the presence of carbon remnants, which explains the brown color in the image of Trial B

shown in Figure 6. Trial C took precedence over Trial A because of the minimal weight loss before 600 °C was reached, which may have been due to a longer dwell time.

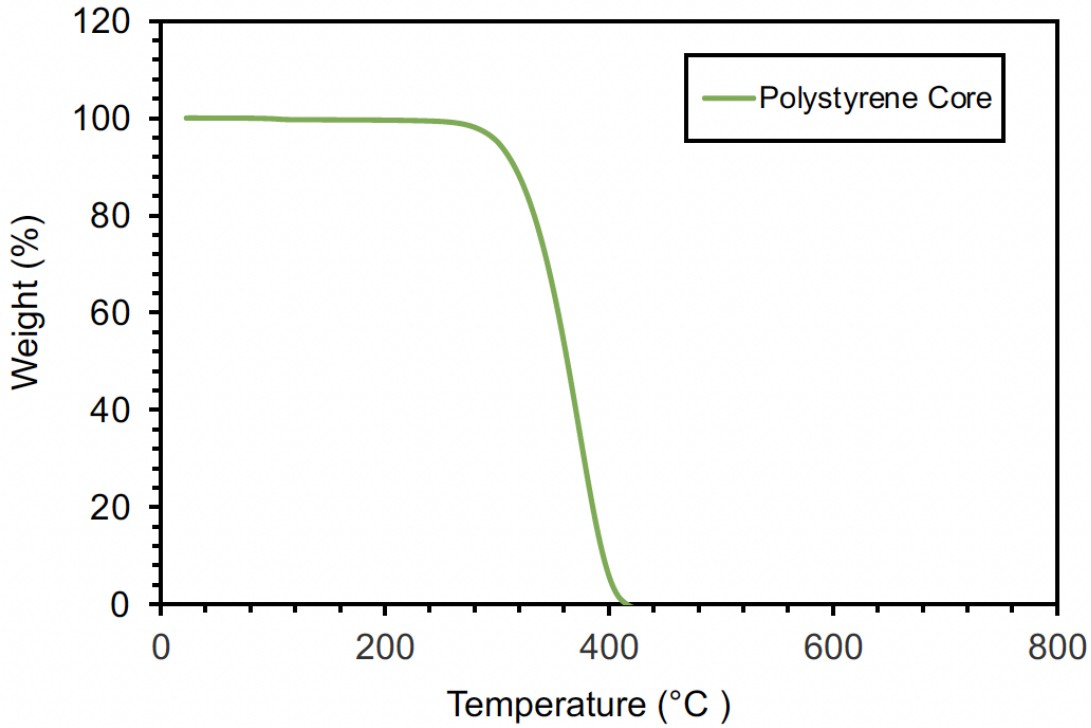

**Figure 5.** Thermal degradation of polystyrene cores using TGA.

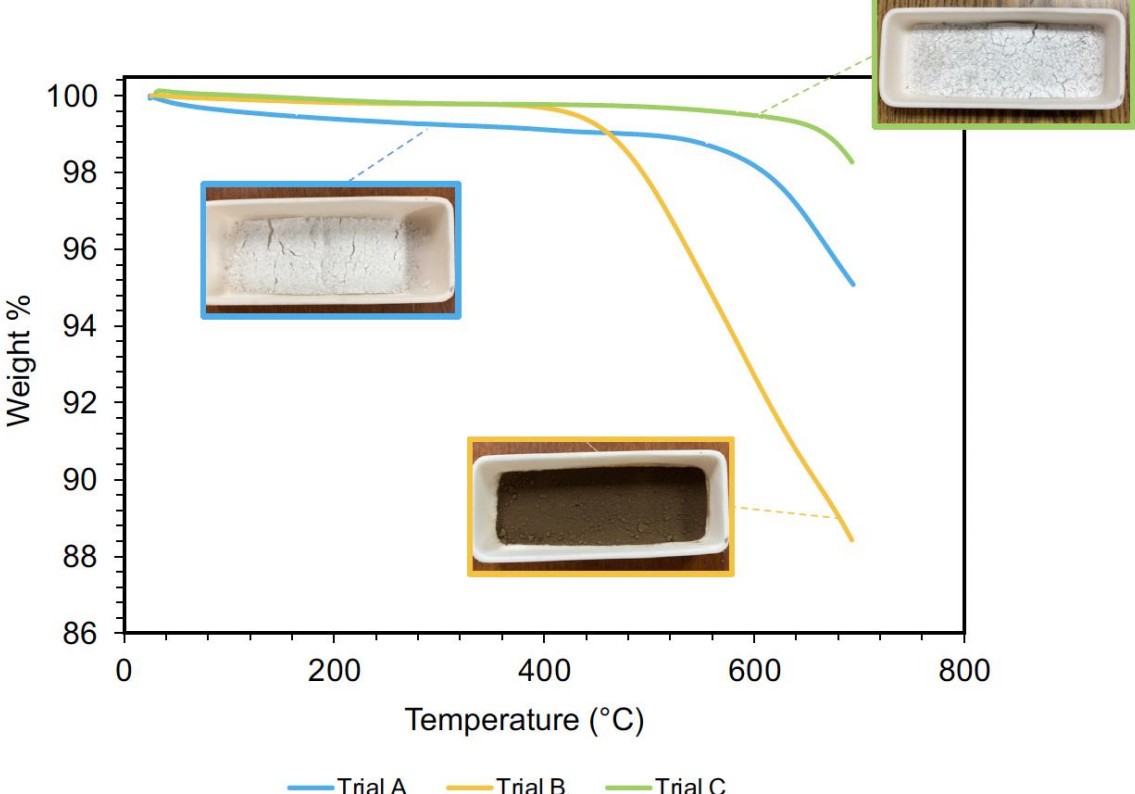

**Figure 6.** TGA of polystyrene–polysiloxane composites using different calcination protocols. Visual representation of hollow polysiloxane microspheres following calcination.

SEM analysis, shown in Figure 7, determined whether there was breakage due to a slightly higher ramp rate of 4 °C/min versus 1 °C/min. Figure 7 shows the SEM analysis that was conducted to visualize the calcination process and observe the hollowness of the polysiloxane shells at different calcination temperatures and with varying durations. Figure 7 also displays the spectra of each of the three calcinated samples using FTIR to identify the presence of residual polystyrene left inside the shell. Trial A and Trial C represent effective etching protocols at a higher dwell temperature of 500 °C. FTIR results were analyzed to display active bonds within the calcinated composites and pure polystyrene spectra were analyzed for the comparison of these active bonds. A core–shell spectrum before calcination was also carried out to consider the effects of sol–gel polymerization on polystyrene characteristic bonds. Following the trend of the TGA and SEM results, observation was centered at the Trial B spectra, where bands associated with C=C bonds appear at 1600 cm$^{-1}$ and a sharp peak indicating C-H out-of-plane bending appears at 700 cm$^{-1}$ [39].

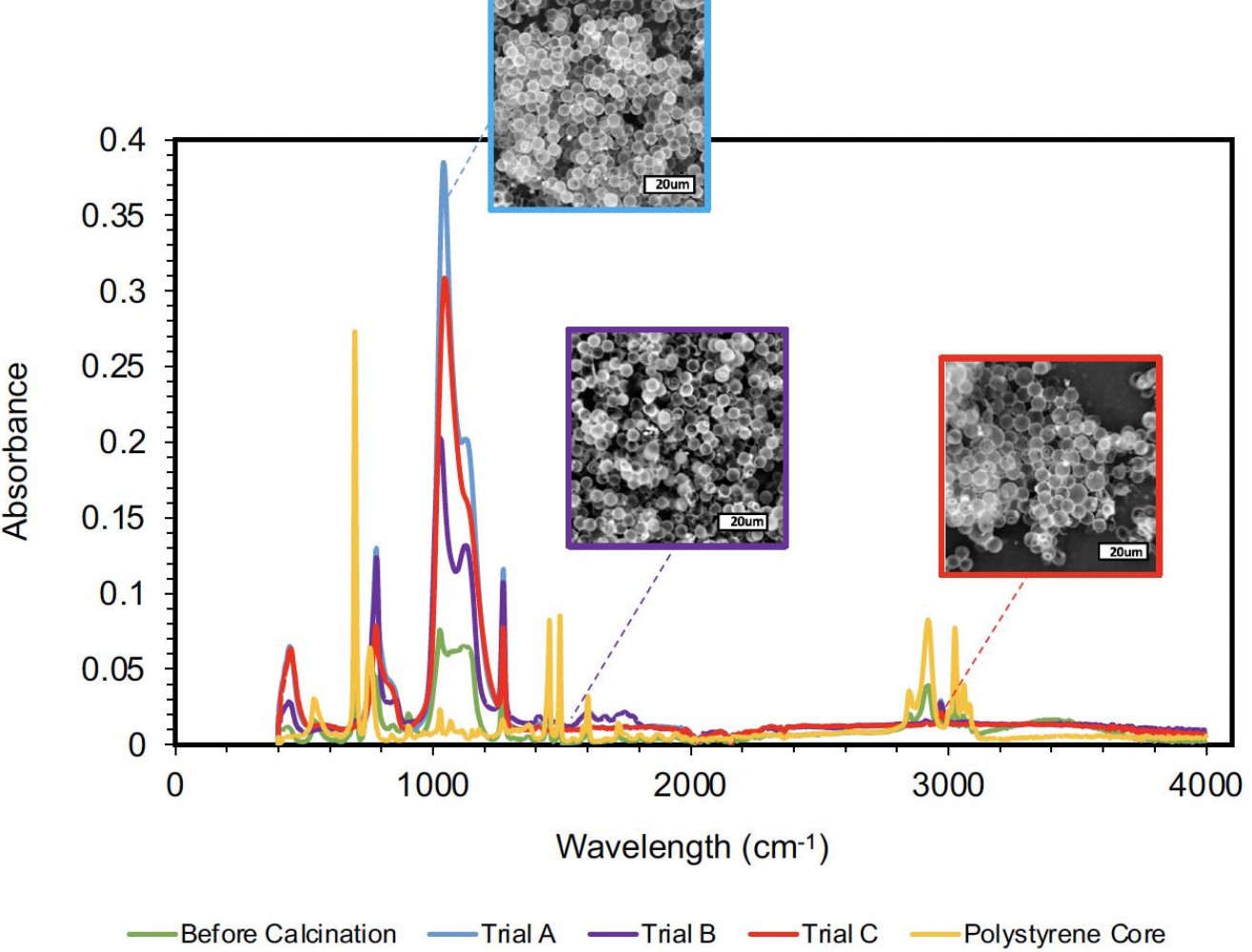

**Figure 7.** FTIR spectroscopy of calcinated particles, core–shell composites with no calcination, and polystyrene cores. Visual comparison of calcination trials A, B, and C.

### 4.4. PDMS–Polysiloxane Syntactic Foams

Figure 8 depicts a cross-section of a PDMS syntactic foam before, after, and during in situ compression. Polymeric syntactic foams with hollow polysiloxane microspheres were fabricated using traditional mold casting methods (Figure 8D). The HPMs that were used for syntactic foam fabrication were from Trial C (Figure 3). Hollow polysiloxane microspheres were integrated into the PDMS to take up an approximately 50 vol% and

70 vol% of the syntactic foam. In Figure 8B, hollow polysiloxane microspheres appeared to deform at 50% uniaxial compression. This confirmed the distribution of compressive stress across the matrix and the HPMs. We observed the cross-section of the syntactic foam before (Figure 8A) and after compression (Figure 8C), indicating excellent shape recovery and elasticity after the compressive force was removed.

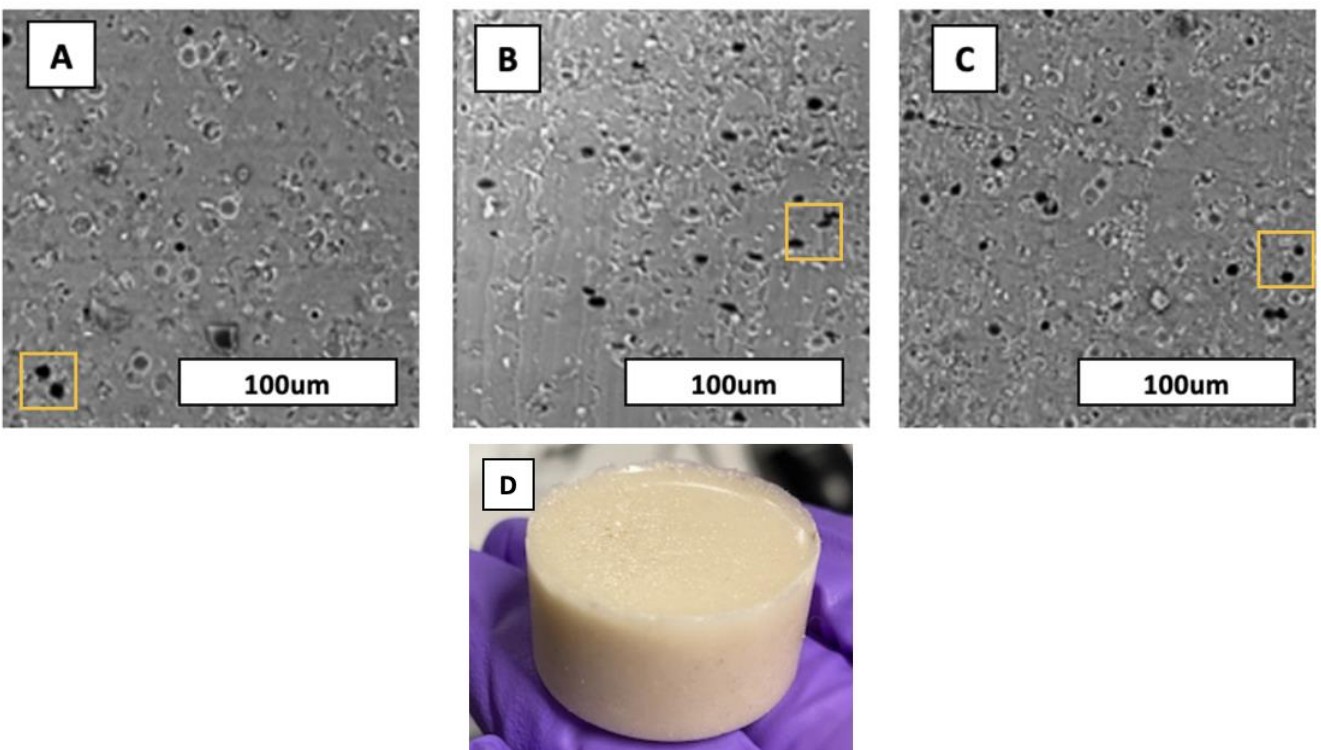

**Figure 8.** SEM of a cross-section of syntactic foam (**A**) before compression, (**B**) under in situ compression, and (**C**) after compression. (**D**) A macroscopic image of the polymer syntactic foam.

*4.5. Energy Absorption and Elastic Recovery of Syntactic Foams*

4.5.1. Macroscopic Compression Comparison

To begin the investigation of how hollow polysiloxane microspheres impact a PDMS matrix in terms of strength and energy absorption, we fabricated a 50 vol% syntactic foam. Figure 9 displays the stress–strain curves for the syntactic foams, bulk PDMS, and two porous PDMS specimens. The porous PDMS specimens were identified as PDMS 1, with about 30% porosity and a monodispersed pore distribution, and PDMS 2, with about 50% porosity and polydisperse pore distribution. Porous PDMS specimens were fabricated through emulsion induction.

Compression of 50 vol% polymer syntactic foam showed the distinguished behavior of an elastomeric foam, which includes three distinct regions on the stress–strain curve. The three main regions used to analyze polymeric syntactic foam compression response were as follows: the initial slope (Young's modulus), the plateau/middle region, and final densification [21]. The initial slope was directly impacted by the stiffness of the PDMS matrix material. The plateau region was where a majority of the energy was absorbed in the composite foam. Figure 8B shows that this was when the hollow polysiloxane microspheres began to deform. Complete densification happened when the hollow microsphere walls collapsed and closed the "pore", imitating the composition of a fully dense specimen. In the case of the fabricated 50 vol% PDMS syntactic foam with hollow polysiloxane microspheres, the initial densification was taken from a strain of 0–0.05 mm/mm. The plateau region began at 0.051 mm/mm and ended at 0.55 mm/mm strain. Finally, the complete densification of the PDMS–polysiloxane composite material was observed at a

strain beyond 0.56 mm/mm. Porous PDMS specimens and bulk PDMS material did not exhibit the distinct mechanical characteristics of the 50 vol% syntactic foam; however, for the sake of direct comparison, the mechanical properties of each specimen were quantified in the same regions. In the case of qualitative analysis, the stress–strain curves for the porous PDMS specimens with no hollow particle reinforcements underwent failure after reaching a strain of 0.5 mm/mm. Bulk PDMS specimens also experienced some fracture after 0.65 mm/mm strain. To quantify the energy absorbed for each specimen, the following equation was used [40]:

$$U = \int_0^{\varepsilon_{max}} \sigma \, d\varepsilon \, \frac{J}{mm^3} \tag{1}$$

The initial densification of the 50 vol% PDMS syntactic foam was two times greater than that of the bulk specimen and three times greater than porous PDMS 2 (a polydisperse system). In terms of Table 1, it was apparent that the Young's modulus of PDMS was directly correlated to energy absorption capabilities in the sense that a higher modulus was associated with higher energy absorption. The integration of HPMs into the PDMS matrix resulted in a roughly 120% increase in energy absorbed, when compared with a bulk PDMS specimen, and a 104% increase in modulus. Compressive strength also increased in the presence of hollow microspheres in the polymer matrix, where the 50 vol% syntactic foam exhibited an increase in strength of about 20% when incorporated into PDMS.

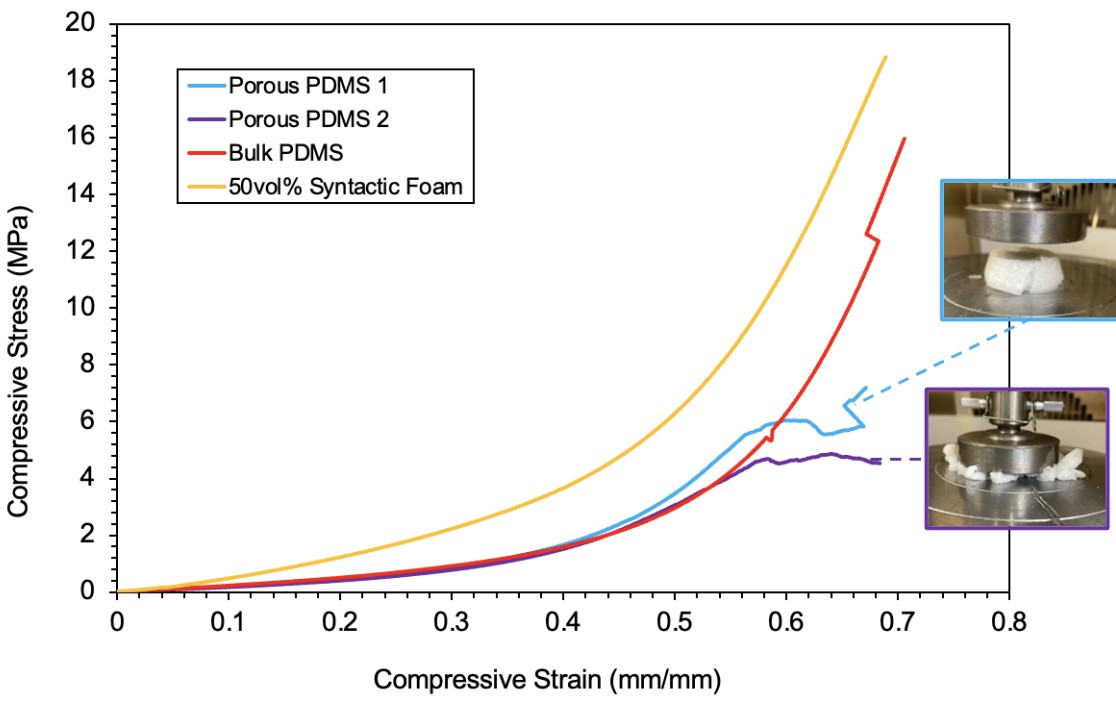

**Figure 9.** Macroscopic uniaxial compression of PDMS in different ways: bulk, porous, and 50 vol% syntactic foam formulations.

**Table 1.** Mechanical response of porous PDMS, bulk PDMS, and 50 vol% syntactic foam specimens.

| Specimen Type | Young's Modulus | Energy Absorption (Plateau Region) | Ultimate Compressive Strength |
|---|---|---|---|
| Porous PDMS 1 | 1.612 MPa | 0.748 J/mm$^3$ | 7.21 MPa |
| Porous PDMS 2 | 1.315 MPa | 0.652 J/mm$^3$ | 4.54 MPa |
| Bulk PDMS | 1.749 MPa | 0.684 J/mm$^3$ | 15.9 MPa |
| 50 vol% Syntactic Foam | 3.575 MPa | 1.505 J/mm$^3$ | 18.9 MPa |

4.5.2. Uniaxial Cyclic Compression of Syntactic Foams at 70% Compression

Section 4.5.1 shows the performance of hollow polysiloxane microspheres and demonstrates that there were considerable energy absorption capabilities when 50 vol% syntactic foams were fabricated. In this next section, we detail the use of cyclic compression to reveal the Mullins effect and determine how hollow polysiloxane microspheres contributed to the stress relaxation phenomena observed in vulcanized rubbers and elastomers. In this case, 70 %vol PDMS syntactic foams were compressed to also identify the effects of incorporating more hollow spheres. Three compressions took place, with a five-minute rest in between each compression. The stress–strain curves of cyclic compression can be seen in Figure 10 below.

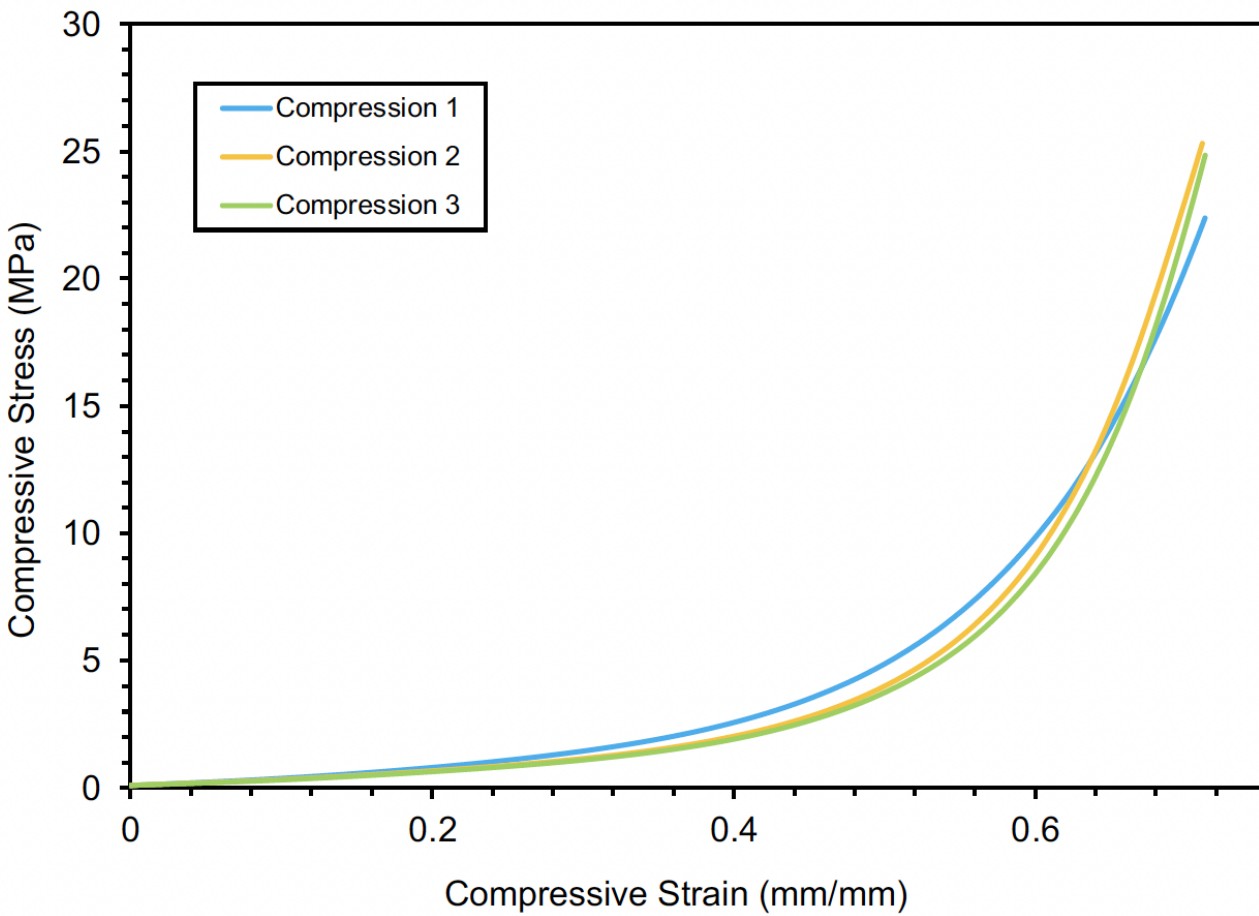

**Figure 10.** Visualization of the macroscopic uniaxial compression of 70 vol% syntactic foams to demonstrate the Mullins effect.

Uniaxial testing was conducted at 70% compression for 70 vol% PDMS–polysiloxane syntactic foams. In Table 2, the Young's modulus of the syntactic foam dropped by 20% from compression 1 to 2 and 9% from compression 2 to 3. It can be deduced that the drop in modulus after loading was due to softening of the PDMS matrix, and the initial drop was the largest reduction in elasticity. When observing the plateau area at the ranges of 0.051 mm/mm strain to 0.6 mm/mm strain, stress softening was prevalent. Following the first compression, the syntactic foams experienced the effects of viscoelastic characteristics. Compressive stresses that were exerted on the syntactic foam were now reached at higher strains after each cycle. In Figure 10, the final densification following 0.6 mm/mm strain underwent a stiffening after the first compression and then relaxed following the second compression. The initial stiffening may have been due to the breakage of hollow polysiloxane microspheres that were embedded within the PDMS matrix and attributed to higher compressive strength [21]. Comparing the energy absorbed and the

Young's modulus of 50 vol% and 70 vol% polymer syntactic foams, the 50 vol% was superior over all specimens tested in both cases. Interestingly, a higher volume percentage of HPMs did not correlate to enhanced mechanical performance. The extra 20 vol% of hollow microspheres resulted in less PDMS area between spheres and possibly induced more strain on them that resulted in further breakage. When comparing the energy absorbed from 50 vol% and 70 vol% syntactic foams, a 40% decrease in the 70 vol% foam was observed. The Young's modulus of both syntactic foam specimens followed a similar trend, where there was a 3% decrease due to a higher microsphere volume percentage. However, ultimate compressive strength did increase by 34% with increasing HPM presence, as shown in Table 2.

**Table 2.** Mechanical response of PDMS–polysiloxane 70 vol% syntactic foam specimens during cyclic loading.

| Compression # | Young's Modulus | Energy Absorption (Plateau Region) | Ultimate Compressive Strength |
|:---:|:---:|:---:|:---:|
| 1 | 2.569 MPa | 1.461 J/mm$^3$ | 22.374 MPa |
| 2 | 2.131 MPa | 0.844 J/mm$^3$ | 25.308 MPa |
| 3 | 1.959 MPa | 0.642 J/mm$^3$ | 24.841 MPa |

4.5.3. Uniaxial Cyclic Compression Comparison

Comparison between multiple modes of elastomeric materials was beneficial for proper selection based on low-stress or high-strain applications. In Figure 10, a 50 vol% syntactic foam, bulk PDMS, and monodisperse porous PDMS were subjected to cyclic loading. Bulk PDMS specimens were considered as a baseline for our elastomeric foams. In Figure 11, upon observation of the stress–strain curves, it was apparent that there was a dramatic increase in modulus throughout the syntactic foam specimens in comparison to the other specimens. The increase in syntactic foam stiffness could be due to polysiloxane exhibiting a high Young's modulus after high thermal exposure, as seen when generating polymer-derived ceramics [41]. In our case, the polysiloxane microspheres experienced calcination at 500 °C; thus, it is possible that there may have been additional cross-linking within silanol groups that contributed to microsphere stiffness. When we observed the mechanical behavior of the 50 vol% syntactic foam after the first compression, there was an intense drop in modulus and a greater densification slope after a compressive strain of 0.45 mm/mm. After the first compression for the 50 vol% syntactic foam, stress softening was observed from 0.2 mm/mm to 0.45 mm/mm. After the second compression, stress softening continued during the plateau region and there was a stiffening at the final densification region. The bulk PDMS specimens behaved in a somewhat linear fashion between compressive stress and compressive strain and did not experience distinct initial stiffness or final densification. The porous PDMS specimen tested followed similar curve behavior to the syntactic foams, where there was a plateau and then a final densification region. Although the curve for porous PDMS followed the same trend as that of the syntactic foam, their energy absorption capabilities dwindled as they reached higher strains. With cyclic loading, porous PDMS stress softening was not as obvious as the behavior of the syntactic foams in this case.

In Table 3, the results for energy absorption at low stresses and high strains for all samples were compiled for each compression. The compressions of each specimen were annotated as C1, C2, and C3 to indicate compression 1, compression 2, and compression 3, respectively. Porous PDMS specimens were favorable over 50 vol% syntactic foams in terms of low-stress energy absorption capabilities. The porous samples had a 170% increase, on average, in energy absorbed at the low stress of 0.5 MPa when compared to the polymeric syntactic foams. When considering the energy absorbed overall, 50 vol% syntactic foams outperformed bulk PDMS specimens by 520% and porous PDMS specimens by 447%. The total energy absorbed at 50% compressive strain of the 50 vol% syntactic foams followed the same downward trend as the 70 vol% sample (Figure 10). The energy absorption of

the 50 vol% syntactic foam dwindled by 7% after the first compression and by 3% after the second compression. The standard deviation between all energy absorption values for the syntactic foam is 0.049. However, the same trend is not followed by the bulk and porous PDMS 1 specimens. In contrast, there was only a 0.004 and 0.005 standard deviation in the bulk specimen and porous PDMS 1 energy absorption values, respectively.

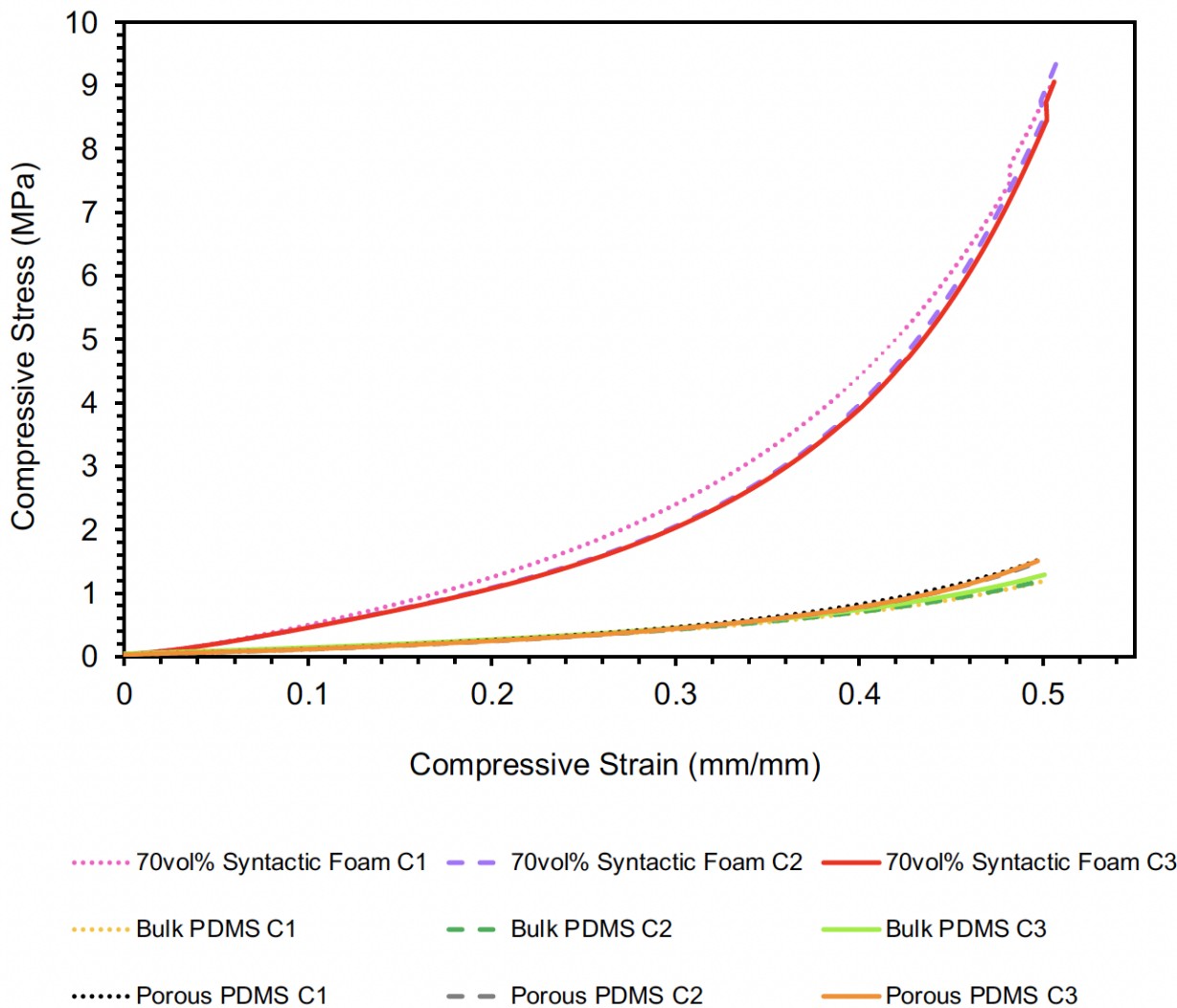

**Figure 11.** Macroscopic cyclic compression of 50 vol% syntactic foams, bulk PDMS, and 30% porous PDMS at 50% compression.

**Table 3.** Mechanical response of PDMS specimens.

| Specimen Type | Energy Absorption at $\sigma = 0.5$ MPa | Energy Absorption at $\varepsilon = 0.5$ mm/mm |
|---|---|---|
| 50 vol% Syntactic Foam C1 | 0.024 J/mm$^3$ | 1.279 J/mm$^3$ |
| 50 vol% Syntactic Foam C2 | 0.026 J/mm$^3$ | 1.193 J/mm$^3$ |
| 50 vol% Syntactic Foam C3 | 0.027 J/mm$^3$ | 1.162 J/mm$^3$ |
| Bulk PDMS C1 | 0.077 J/mm$^3$ | 0.207 J/mm$^3$ |
| Bulk PDMS C2 | 0.076 J/mm$^3$ | 0.211 J/mm$^3$ |
| Bulk PDMS C3 | 0.076 J/mm$^3$ | 0.220 J/mm$^3$ |
| Porous PDMS C1 | 0.069 J/mm$^3$ | 0.234 J/mm$^3$ |
| Porous PDMS C2 | 0.070 J/mm$^3$ | 0.224 J/mm$^3$ |
| Porous PDMS C3 | 0.071 J/mm$^3$ | 0.225 J/mm$^3$ |

#### 4.5.4. Time Recovery following Cyclic Compression Tests

Elastic hysteresis is a stress relaxation event that elastomers often experience after cyclic loading [42]. In this work, we hypothesized that the hollow polymer microspheres would contribute to elastic recovery within the polymer matrix. After initial cyclic loading, the samples were stored in an ambient environment. After a one-week recovery period, the three samples were subjected to cyclic compression, and shape recovery was directly measured by specimen height. In Table 4, the dimensions reflect the syntactic foam's ability to retain its original height before loading. Three specimens of the same type (50 vol% syntactic foam) were fabricated and tested to ensure consistency. The first specimen recovered about 99% of its original height, the second specimen recovered 100% of its original height, and specimen 3 recovered approximately 99% of its original height. There was also a difference between the differences in heights after the first and second compressions, thus suggesting permanent deformation from increasing load cycles.

**Table 4.** Dimensions of 50 vol% polymer syntactic foams before compression, immediately after compression, after 1-week recovery, and after the second set of cyclic compressions.

| Height | Specimen 1 | Specimen 2 | Specimen 3 |
| --- | --- | --- | --- |
| Original | 12.41 mm | 12.11 mm | 12.10 mm |
| After 1st compressions | 12.34 mm | 12.02 mm | 11.99 mm |
| After 1 week of recovery | 12.4 mm | 12.11 mm | 12.08 mm |
| After 2nd compressions | 12.31 mm | 11.97 mm | 11.95 mm |

The work showcased illustrates that new knowledge was brought forth on the fabrication of polystyrene microspheres, hollow polysiloxane microspheres, and polymer syntactic foam composites. Lu et al. demonstrated that the sol–gel process discussed in their work yielded a maximum diameter of 3 um [16], whereas we have discussed implementations that led to the synthesis of polystyrene with a diameter of 11 μm and low population variation. Tewani et al. studied the mechanics of polymer syntactic foams by varying glass micro balloon diameters and found that larger diameter hollow microspheres led to an increase in strength [43]. Yousef et al. reported their work on syntactic foam compressive response for 2%, 10%, and 40% microsphere inclusion and discovered that the Young's modulus increased with increasing microsphere percentage [21]. In the work presented, 50% and 70% volumes of hollow microspheres in a polymer matrix were mechanically tested. It was observed that the Young's modulus is not necessarily linear with increased hollow microsphere presence; however, it was noted that hollow polymer microspheres influence the mechanical strength of syntactic foam properties in compression.

#### 5. Conclusions

The optimization of polystyrene cores through variable reaction temperatures, the implementation of different sol–gel protocols, and altered calcination procedures was important to yield hollow polysiloxane microspheres. Hollow polysiloxane microspheres were synthesized for implementation into a PDMS matrix to enhance its mechanical properties. Uniaxial macroscopic compression tests were performed to quantify the compressive strength, Young's modulus, and energy absorption of polymer syntactic foams with a 50 vol% and 70 vol% of HPMs. Comparisons were made between varying hollow particle volume dispersions in syntactic foams, as well as bulk and porous PDMS samples with no hollow particle reinforcements. We found that the inclusion of microspheres was not linearly correlated with enhancements to mechanical properties, rather, different volume percentages yielded different results. In the case of 50 vol% syntactic foams, the inclusion of hollow polysiloxane microspheres considerably increased energy absorption by up to 520%. In contrast, the increase in the presence of HPMs in the polymer matrix to achieve a 70 vol% syntactic foam increased its compressive strength by 34%. At higher strains, polymer syntactic foams can withstand failure due to the energy absorbed, which was

facilitated by the presence of elastic hollow microspheres. The polymer syntactic foams also exhibited superior shape memory and can withstand multiple compressions while still absorbing energy after each cycle. In future work, it would be beneficial to mechanically test individual hollow microspheres to understand their elasticity and material strengths.

**Author Contributions:** Conceptualization, S.G.G. and A.I.; methodology, S.G.G. and A.I.; software, S.G.G. and K.E.-R.; validation, S.G.G., A.I. and Y.L.; formal analysis, S.G.G. and A.I.; investigation, S.G.G. and S.G.; resources, Y.L. and E.S.; data curation, S.G.G.; writing—original draft preparation, S.G.G. and S.G.; writing—review and editing, S.Z., Y.L., and M.S.H.; visualization, S.G.G.; supervision Y.L.; project administration, Y.L.; funding acquisition, Y.L. All authors have read and agreed to the published version of the manuscript.

**Funding:** This research was funded by grant number DE-NA-0004051.

**Data Availability Statement:** Data is not available due to patent pending intellectual property.

**Conflicts of Interest:** The authors declare no conflict of interest.

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
