# Peer review of "Fabrication and Characterization of Hollow Polysiloxane Microsphere Polymer Matrix Composites with Improved Energy Absorption"

_jcs, doi:10.3390/jcs7030098_

Round 1

Reviewer 1 Report

This manuscript reported a hollow polysiloxane microsphere and PDMS composites, achieving superior energy absorption. Overall, this topic is interesting and meaningful, and the manuscript is also logical for reading. However, there are some concerns and suggestions which, if addressed, would enhance the quality of the manuscript. These are provided below:

1. In Figure 4d, it is difficult to observe that the polysiloxane has been fully coated on the surface of polystyrene microsphere.

2. In Figure 7, the authors said “SEM analysis for Trial B displays several opaque spheres, indicating the presence of a polystyrene core”, the reviewer think it is hard to convince. Because the SEM can only observe the surface morphology of microsphere, but can not see the internal morphology of the microsphere, unless the ball is broken or uses the TEM. As for the opaque spheres or transparent spheres, the reviewer believe it is only connected with the testing condition.

3. Besides, this manuscript needs careful editing in technical English editing. And many Figures are not well prepared, it is suggested to give normative images by using “Origin” software.

Reviewer 2 Report

please, see the attachment

Round 2

Reviewer 1 Report

The authors have done careful revising according to the comments, I suggest it can be accepted in present form.

Reviewer 2 Report

All major comments were adequately addressed and the Authors have done an admirable job of improving the quality of the manuscript. Therefore, it can be accepted without any structural modification.